# Sequesterpene Lactones Isolated from a Brazilian Cerrado Plant (*Eremanthus* spp.) as Anti-Proliferative Compounds, Characterized by Functional and Proteomic Analysis, Are Candidates for New Therapeutics in Glioblastoma

**DOI:** 10.3390/ijms21134713

**Published:** 2020-07-01

**Authors:** Clarice Izumi, Helen Julie Laure, Nayara Gonçalves Barbosa, Carolina Hassibe Thomé, Germano Aguiar Ferreira, João Paulo Barreto Sousa, Norberto Peporine Lopes, José César Rosa

**Affiliations:** 1Protein Chemistry Center and Department of Cell and Molecular Biology and Pathogenic Bioagents, Ribeirão Preto Medical School, University of São Paulo, Ribeirão Preto 14049-900, SP, Brazil; helen_julie_l@yahoo.com.br (H.J.L.); nbarbosa@usp.br (N.G.B.); jcrosa@fmrp.usp.br (J.C.R.); 2Department of Biochemistry and Immunology, Ribeirão Preto Medical School, University of São Paulo, Ribeirão Preto 14040-900, SP, Brazil; carolhthome@gmail.com (C.H.T.); germanoaf@usp.br (G.A.F.); 3Núcleo de Pesquisas de Produtos Naturais e Sintéticos, School of Pharmaceutical Sciences of Ribeirão Preto, University of São Paulo, Ribeirão Preto 14040-903, SP, Brazil; jpsousa@fcfrp.usp.br (J.P.B.S.); npelopes@fcfrp.usp.br (N.P.L.)

**Keywords:** sesquiterpene lactone, natural products, glioblastoma cells, proteomics, unfolded protein response, temozolomide

## Abstract

Gliomas are responsible for more than 60% of all primary brain tumors. Glioblastoma multiforme (GBM), a grade IV tumor (WHO), is one of the most frequent and malignant gliomas. Despite two decades of advances in the discovery of new markers for GBM, the chemotherapy of choice falls to temozolomide after surgery and radiotherapy, which are not enough to increase the survival of patients to more than 15 months. It is urgent to discover new anti-glioma compounds. Many compounds derived from natural products have been used in the development of anti-tumor drugs. In this work, we have screened six low molecular weight sesquiterpene lactones, isolated from *Eremanthus* spp., and studied their function as anti-proliferative agents against GBM strains. We demonstrated that two of them, goyazensolide and lychnofolide, were effective in reducing cell viability, preventing the formation of anchorage-dependent colony and were able to pass through a mimetic blood-brain barrier making them candidates for glioma therapy, being more potent than temozolomide, according to in vitro assays for the cell lines tested. Proteomic analysis revealed a number of altered proteins involved in glycolytic metabolism and cellular catabolism.

## 1. Introduction

Gliomas are responsible for more than 60% of all primary brain tumors. Glioblastoma multiforme (GBM), a grade IV tumor (WHO), is one of the most frequent and malignant gliomas [1,2]. Temozolamide (TMZ) is the only drug of first choice for glioma chemotherapy after surgery and radiation therapy. However, due primarily to the over-expression of O6-methylguanine methyltransferase (MGMT) and/or lack of a DNA repair pathway in GBM cells, some patients do not respond to TMZ [3].

*Eremanthus* has been considered one of the most numerous genera of subtribe Lychnophorinae, which belongs to the Vernonieae tribe of the Asteraceae family [4]. The *Eremanthus* plant species are widely distributed throughout mountain ranges in the Central and Southeastern regions of Brazil, especially in the states of Minas Gerais, Bahia and Goiás, and have been extensively studied because of their anti-inflammatory and analgesic activities, attributed to the sesquiterpene lactones of the goyazensolide moiety [4,5]. Additionally, the goyazensolide and its derivatives have proved to be potentially cytotoxic against different tumoral adherent (human colon, breast, glioma, and prostate) and non-adherent cell lines (human leukemia) [6].

In the present study, we compared the action of the six sesquiterpene lactones isolated from leaves or branch extracts of *Eremanthus seidelii* or *Eremanthus matogressensis* to other drugs, such as thapsigargin (THP), tunicamycin (TUN) and temozolomide (TMZ), on cellular stress and cytotoxicity. We have combined the use of functional assays such as cell proliferation, clonogenicity, cell membrane permeability, cell migration, and proteomic profiling based on microarray antibodies and mass spectrometry to elucidate the mechanisms of action and provide knowledge of these natural compounds in their proposal as candidates for glioma therapy.

## 2. Results

### 2.1. Anti-Proliferative Activities of Six Sesquiterpene Lactones Isolated from Eremanthus spp.

In the present work, we investigated the anti-proliferative properties of six sesquiterpene lactones isolated from *Eremanthus* spp. The compounds were named AM01, AM02, AM03, AM04, AM05 and AM06; their definitions of chemical names are demonstrated in the Section 4 and for ease we use this simplified designation. The sesquiterpene lactone compounds were added to cell cultures at concentrations of 10, 50 and 100 μM. The control was 1% dimethyl sulfoxide (DMSO). Since the compounds have similar molecular mass, the results obtained can be directly compared. It can be seen that compounds AM01 (Figure 1A) and AM03 (Figure 1C) were not effective to prevent cell proliferation in both cell lines, U87MG and T98G. While, compound AM02 (Figure 1B) was selective against the U87MG lineage. AM06 (Figure 1F) demonstrated a dose dependent response for both cell lines, but when compared to the treatment observed with AM04 (Figure 1D) and AM05 (Figure 1E) they demonstrated effectiveness from 10 μM. Thus, we determined that compounds AM04 and AM05 could be candidates for anti-neoplastic therapies, at least in vitro against two cell lines representing glioblastoma, but with a distinct genetic background as discussed later.

### 2.2. Clonogenecity Activities of Six Sesquiterpene Lactones from Eremanthus spp.

The clonogenic cell survival assay determines the cell’s ability to proliferate indefinitely, thereby retaining its reproductive capability to form a large colony or a clone. Although having different plating efficiencies, T98G (31.9%) and U87MG (1.8%), the survival fractions (SF) of the cells treated with the different compounds were equivalent for the two cell lines. The clonogenic assay shown in Figure 2 (panel A, U87MG, and panel B, T98G) demonstrated that at the highest dose, 2000 nM, all compounds tested had some effect on the cell lines, however, AM04 and AM05 were more effective than others at preventing the formation of colony at dose of 500 nM, drastically reducing the survival fraction (SF).

The plating efficiency (PE) for T98G cells was 31.9%, whereas for U87MG cells it was 1.8%. At the concentration of 2000 nM, it reduced the formation of colonies for drugs AM01, AM02, AM03 and AM06 by around 50%. While for the most effective drugs, AM04 and AM05, at this concentration, there was complete prevention in the formation of colonies. Based on the results of the clonogenic assay the compounds AM04 and AM05 demonstrated greatest efficiency in preventing colony formation. Therefore, AM04 and AM05 were tested for transposition efficiency through membrane in a mimetic blood-brain barrier assay using Madin-Darby canine kidney (MDCK) cell lines in transwell-type culture plates. The passage through the membrane is one of the limiting factors for the effectiveness of therapeutic drugs in the treatment of astrocytomas and other gliomas. To the best of our knowledge, only TMZ has been used in the clinic because of its low toxicity and ability to reach the tumor site.

### 2.3. Transwell Assay to Mimic the Passage of Compounds by Mimetic Blood-Brain Barrier Using MDR-MDCK Cells

The capacity of compounds AM04 and AM05 was tested to overcome membranes such as the blood-brain barrier. For this we carried out a transwell assay, where inserts contained a monolayer of MDR-MDCK cell, a cell type that presents intercellular adhesion forming an apical and a basal face. Compounds AM04, AM05 and TMZ were added at the concentration of 5, 15 and 2000 μM in the transwell plate respectively, without insert (MDCK-) and with insert (MDCK+). The results are shown in Figure 3. U87MG were cultured for 24 h in the basal chamber, in which the medium containing the compounds was directly added, MDCK-, while in wells with the inserts MDCK(+), the medium with compounds were placed into the inserts. The MDCK- and MDCK+ wells were subjected to an additional 24 hours incubation as well as control of untreated U87MG cells. The assays were performed in triplicates and reported as mean ± SEM. Compounds AM04 (paired one-tailed distribution Student t-Test *p* = 0.0027) and AM05 (*p* = 0.0046) showed comparable efficiency to cross the blood-brain mimic barrier membrane as TMZ (*p* = 0.0003), with a concentration of 133 to 200 times lower than TMZ under the same assay condition, respectively. It should be noted that the compounds subjected to the insert with MDCK cells had an approximate 6-fold dilution relative to the lower chamber (100 µL in the insert and 600 µL in the basal compartment) and may explain the relative minor effect on the viability of U87MG cells due to this dilution. The results obtained for cell viability were statistically significant (*p* < 0.01). At these concentrations, the compounds used did not affect the viability of MDCK cells (see Appendix A). Also, for TMZ, there was no impairment of MDCK cells that had already reached confluence and were no longer replicating, thus not being targeted for TMZ, an alkylating DNA agent. To determine the integrity of the mimic membrane, we used Lucifer Yellow, in parallel well, and we observed that no fluorescence was detected in the lower chambers when MDCK+ insert was used.

### 2.4. Wound Healing Assay

To detect the effect of the different drug treatments on glioma cells migration and invasion, we performed the wound healing assay. Wound healing assay results showed that migration of U87MG and T98G cells was significantly decreased by THP and TUN treatments compared to control (Figure 4), having TUN treatment likely caused cell death. No differences were observed for AM04 (goyazensolide) and AM05 (lychnofolide) compared to untreated cells. No difference was observed between the treatments. These results suggest that TUN and THP could decrease mobility and invasiveness in glioma cancer cells.

### 2.5. Proteome Profiler Human Apoptosis and Stress Antibody Arrays

Next, we submitted the protein extracts of the GBM, U87MG and T98G to membrane-based antibody array for the parallel determination of the relative levels of 35 human apoptosis-related proteins. Proteome profiling assay had highlighted the involvement of several apoptotic proteins in promoting apoptosis in treated U87MG and T98G with compounds AM04, AM05 and TMZ at 10 µM, 15 µM and 2 mM, respectively. The intensity of duplicates of each spot (antibody) was measured by densitometry and calculated as a percentage relative to untreated cells and placed on a color-guided heat-map as shown in Figure 5. The results of each of the arrays are shown in Appendix A for apoptosis assay and Appendix A for cell stress assay, respectively. Figure 5 demonstrates that apoptosis was induced at high intensity for the U87MG lineage and at lower intensity for T98G for TMZ. However, treatment with AM04 and AM05 did not significantly induce apoptosis in GBM cells. Only for U87MG was there an increase in detection for cleaved CASP3 and TNFR1/TNFRSF1A with the compounds AM04 and AM05, but the same effect was not observed for T98G. In conclusion, only TMZ was able to induce apoptosis in both cells by proteins included in intrinsic and extrinsic apoptotic pathways.

Next, we submitted the same protein extracts of the GBM, U87MG and T98G to membrane-based antibody array for the parallel determination of the relative levels of 26 human cell stress-related proteins, using the same method employed for the proteins related to apoptosis. We submitted 26 proteins to GO-TermFinder for enrichment in biological processes (*p* < 0.01), resulting in 12 proteins divided into three main categories: response to oxidative stress (7/26); cell proliferation (8/26) and cell death in response to oxidative stress (3/26), which are highlighted in Figure 6, also distributed in a color-guided heat-map. We investigated through proteome profiler proteins related to cell stress, although the method is semi-quantitative, data acquisition and densitometric readings were performed at the same time in the set of treatments and compared to untreated cells. Regarding treatment with AM04 and AM05, we can note that the up-and down-regulation effect for the listed proteins were diffused for the treatments with AM04 and AM05, however, for TMZ there was significant up-regulation for most proteins, but a smaller effect was observed for U87MG. It should be noted that HSBP1 with chaperone function and Thireodoxin (TXN) with anti-oxidant function [7] were up-regulated in both strains by TMZ, but not by AM04 and AM05. It is noteworthy that two proteins in AM04 and AM05, ADAMTS1 and p27 (KIP) were up-regulated and related to cell proliferation, except in TMZ treated cells, where p27 (KIP), a cyclin-CDK inhibitor, was down-regulated. In conclusion, these results indicate that TMZ preferentially induces apoptosis in response to DNA damage, whereas sesquiterpene lactone, AM04 and AM05 prevent cell proliferation, probably by increasing p27 expression (KIP), which may indicate this pathway.

### 2.6. Quantitative Proteomic Analysis of GBM Cell Lines by iTRAQ (Isobaric Tags for Relative and Absolute Quantitation)

Subsequently, we used the same protein extracts from proteome profiler apoptosis and stress cell kits to investigate changes in the proteome of GBM strains based on mass spectrometry using iTRAQ when treated with different compounds in order to elucidate a possible mechanism of action for the new AM04 and AM05 compounds. In addition to untreated cell proteome, we extended the research to thapsigargin (THP), tunicamycin (TUN) already widely known to induce unfolded protein response during endoplasmic reticulum cell stress (UPR) and continued comparing to temozolomide (TMZ), the first choice in chemotherapy for gliomas. The proteome changes of the T98G and U87MG lines as a result of the effect of treatment with the compounds are shown in Figure 7. A total of 358 proteins were identified and quantified by iTRAQ in T98G and 283 proteins in U87MG, restricted to 1% FDR and >80% probability for each peptide. Proteins were identified with a minimum of 2 peptides and quantified by a minimum of 3 peptides/protein (Appendix A). Protein differentially expressed is expressed in log 2-fold change and only showed those a statistically significant change (*p* < 0.05) compared to untreated cells.

In T98G there were alterations of GRP78/BIP and HS71A (HSPA1A) proteins in the proteome, which were increased from the treatment effect by AM04 and AM05. Using the GO-TermFinder program (https://go.princeton.edu/cgi-bin/GOTermFinder), these changes were associated with protein refolding (GO: 0042026) and heat shock protein binding (GO: 0031072). Interestingly, cells treated with THP and TUN, known as inducers of unfolded protein response (UPR) with consequent increase in master regulator of UPR, GRP78/BIP, did not show any increase of HS71A (HSPA1A). This detail seems to confer to the compounds AM04 and AM05 a mechanism other than for THP and TUN. In T98G treatment with THP, TUN and TMZ had a common effect on decreasing expression of proteins involved in oxidoreductase activity such as LDHA, FASN, AKR1B10, PRDX6, LDHB and G6PD. This fact allows us to conjecture that the reduction in proteins that act in elimination/prevention of the production of reactive oxygen species should elevate the biological processes involved in cellular stress as well as activate pathways that will lead these cells to cell death. This still draws attention to the significant reduction of mitochondrial protein expression, NQO1 in AM04. What should be highlighted is VDAC1, which was upregulated during treatment with AM04, but not for AM05. VDAC1 was significantly abundant for AM04, THP, TUN and TMZ compared to untreated cells. Perhaps VDAC1 being an important mitochondrial membrane protein deserves to be studied in greater depth in future studies in elucidating the mechanisms of action of these sesquiterpene lactone compounds.

Changes in the U87MG proteome indicated down regulation of the various proteins implicated in NAD biosynthetic process (NAMPT, TPI1, GAPDH, PKM) and (GO: 0061621) canonical glycolysis (TPI1, GAPDH, PKM). This result is totally different from that obtained for T98G. Still, the proteome change was different for the THP, TUN and TMZ compounds, except for UPR master regulator (GRP78/BIP) which increased in both cells, U87MG and T98G. This different effect can be attributed to genetic differences in these strains, where U87MG has a much higher glycolytic potential than T98G [2]. It is possible to speculate that AM04 and AM05 could have a direct action in those types of tumors in which there is a high level of hypoxia and aerobic glycolysis and its pathways are exacerbated (Warburg effect). In U87MG for THP, TUN and TMZ, the drug actions were achieved in the protein metabolic process (GO: 0019538), whereas UPR master regulator GRP78/BIP was increased only in THP and TUN, but not in those treated with TMZ. It is important to remember that the proteomic results were obtained by iTRAQ, in which the samples were mixed after isobaric labeling, which means that in all samples the identification of the proteins is obtained by the fragmentation spectra of CID-MS/MS and do not discriminate the source of peptide structure, while quantification can be individualized for each sample. This means that the identified proteins are the same for all samples, and the reporter ions are expected to indicate statistically significant changes between samples.

In summary, the proteomic analysis results obtained by mass spectrometry were not enough to enrich specific pathways or molecular functions that could contribute to elucidate the mechanisms of action of sesquiterpene lactone compounds, but demonstrated the alteration of important proteins that are responsible for the balance between cellular stress, metabolism and cell death.

## 3. Discussion

In this work, an investigation was carried out on six sesquiterpene lactone isolates from *Eremanthus* spp. collected from the Brazilian cerrado [5]. The anti-proliferative activity indicated that among the six compounds, two compounds, AM04 (goyazensolide, MW 360) and AM05 (lychnofolide, MW 358) were effective in preventing the cell proliferation of two cell types derived from glioblastoma, U87MG and T98G. The choice of these strains fell on their differences in mutations in tumor suppressor genes and oncogenes. The presence of TP53 (mutated) in T98G and not mutated in U87MG gives them a different response to apoptotic stimuli by drugs. Furthermore, T98G is resistant to temozolomide (TMZ) by expressing an MGMT enzyme [3] which is responsible for demethylating 6-methylmethyl-*O*-guanine, exactly the intervention of TMZ to induce DNA damage during cell replication and consequent induction of apoptosis. U87MG does not present exacerbated expression of MGMT, and is therefore sensitive to TMZ. Clinical studies have shown improved survival when a combination of radiation and temozolomide chemotherapy is used, but median survival for GBM patients is still about 15 months [8]. Therefore, there is the urgent need to discover new compounds capable of acting on tumors of the central nervous system and with the ability to pass through the blood-brain barrier. We have demonstrated here that the compounds AM04 (goyazensolide) and AM05 (lychnofolide) succeeded in having an anti-proliferative activity on U87MG by passing through a mimic membrane of blood-brain barrier with the same efficiency of TMZ and at doses 130 to 200 times lower than TMZ. Both AM04 and AM05 did not affect the viability of MDCK cells at the doses used, suggesting that they would have reduced toxicity in non-neoplastic cells. The integrity of the membrane was attested by the retention of lucifer yellow in the MDCK monolayer and the viability of MDCK was confirmed by XTT assay (Appendix A). There are more than 5000 sesquiterpene lactones characterized as secondary metabolites in species of the plant kingdom, in particular in the family Asteraceae [9]. Sesquiterpene lactones used in clinical trials are artemisinin, thapsigargin and parthenolide and many of their synthetic derivatives. These drugs are selective toward tumor and cancer stem cells by targeting specific signaling pathways, which make them lead compounds in cancer clinical trials [10,11].

We have developed research to elucidate possible mechanisms for sesquiterpene lactone compounds by searching for a microarray of antibodies against proteins related to apoptosis and cell stress. While also using the same protein extract from treated versus untreated cells, we used bottom up proteomic approach with isobaric labelling by iTRAQ. When analyzing the set of altered proteins for each treatment by Gene Ontology, we did not obtain enrichment in pathways that could reveal a specific pathway resulting from treatment with AM04, AM05, THP, TUN and TMZ. However, some proteins that have important functions in directing biological processes are highlighted below.

One protein that caught our attention was ADAMTS1. It belongs to a disintegrin and metalloproteinase with thrombospondin motifs (ADAMTS) family and more precisely to the proteoglycanases subgroup based on their common ability to degrade chondroitin sulfate proteoglycans. An important finding is that all compounds tested for T98G and U87MG demonstrated upregulation of ADAMTS1 in the proteome profiler for cellular stress kit. Endogenous ADAMTS proteoglycanases can negatively regulate by sequestration of VEGF, a potent pro-angiogenesis factor, thus affecting the formation of new blood vessels [12]. Such a stimulus mainly focused on AM04 and AM05 to increase ADAMTS1 expression, although speculative, may be an important factor in attacking neoplastic cells with consequent interference in angiogenesis internally and remodeling of the tumor microenvironment externally by the ability to degrade proteoglycans. Another finding worth discussing is the increased expression of p27-KIP1 (CDKN1B) belonging to the family of poorly structured proteins known as Intrinsically Unstructured Proteins (IUP) and has a broad spectrum of activities based on this malleability of its tertiary structure. However, for the work described herein, AM04 and AM05 treatments, but not by TMZ, induced up-regulation of p27(KIP1)/CDKN1B. The activity of p27 indicates a role in the ability to inhibit cyclin A (E)/CDK2, where p27 interacts specifically with cyclin through a KID D1 sub domain inducing a conformational change allowing adjustment in p27 structure to inhibit CDK2, preventing its ability to phosphorylate and consequently induce cell cycle progression [13], such that p27 may play a significant role in the anti-proliferative activity of AM04 and AM05. Obviously, this fact deserves further investigation.

In the bottom up proteomic analysis we could observe that U87MG and T98G responded differently to the biological process revealed by the proteins that were altered. U87MG had decreased expression of proteins involved in glycolytic metabolism, whereas T98G presented a response to cellular catabolism and to drug-induced cellular stress that resulted in increased heat shock proteins such as GRP78/BIP and HS71A, except for TMZ. Heat shock proteins (HSPs) are identified as a set of highly conserved proteins in pro- and eukaryotes. Their expression is upregulated by a variety of cellular stimuli including physical (e.g., heat, irradiation) or chemical stress. Based on their estimated molecular weights and sequence homology, HSPs are classified into different families [14]. In normal conditions, levels of HSPA1A are very low in all cell compartments. This supports folding of nascent polypeptides, prevents protein aggregation, and assists transportation of proteins across membranes and protects cells from lethal damage induced by environmental stress [15]. However, HSPs, primarily GRP78/BIP, have a key role in the control of ER stress and particularly in neoplastic cells, where this proteotoxic stress is brought to higher levels than in normal cells in such a way that it ends up leading the neoplastic cells to a higher survival and consequent resistance to apoptosis [16]. Establishing what would be these proteotoxic limits has been the subject of numerous works, but nothing that has been enlightening to the present study [17]. Thapsigargin, a sesquiterpene lactone from *Thapsia garganica*, is an inhibitor of the sarco/endoplasmic reticulum Ca^2+^-ATPase (SERCA) and raises intracellular Ca^2+^ by blocking the ability of the cell to pump Ca^2+^ into the sarcoplasmic and endoplasmic reticulum inducing an increase in GRP78 expression, and herein was used as a model drug of sesquiterpene lactone. Surprisingly at the doses of THP used, the increase of GRP78 was observed for AM04 and AM05 in T98G, but there was not the same response for U87MG. Previously our group demonstrated that both lines had increased GRP78 expression by THP and that as a consequence there was a decrease in cell migration [2].

## 4. Material and Methods

### 4.1. Cell Lines and Culture

GBM human cell lines (T98G and U87MG), and Madin-Darby canine kidney cells (MDR-MDCK (NBL2)(ATCC^®^ CCL34™)), from now on referred to as MDCK, were obtained from American Type Cell Collection (Manassas, VA, USA). All cells were maintained at 37 °C in 5% CO_2_ in a humidified chamber, as well as for all the experiments conducted with these cells. T98G and U87MG cell lines were cultured in Dulbecco’s modified Eagle’s medium (DMEM) supplemented with 10% heat-inactivated fetal bovine serum (FBS), 10 mM sodium pyruvate, 10.9 mM HEPES, 43 mM sodium bicarbonate, 10 units/mL penicillin and 10 µg/mL streptomycin. MDCK cell line was cultured in minimum essential Eagle’s medium (MEM) containing 2 mM glutamine, 20 mM sodium bicarbonate, 0.1 mM non-essential amino acids (Gibco, NY, USA), 10 mM sodium pyruvate, 10% heat inactivated FBS, 10 units/mL penicillin and 10 μg/ml streptomycin (Gibco, NY, USA).

### 4.2. Chemical Compounds

Six sesquiterpene lactones, isolated from *Eremanthus* spp. [5], were provided by Prof. Norberto P. Lopes (Núcleo de Pesquisas de Produtos Naturais e Sintéticos, School of Pharmaceutical Sciences of Ribeirão Preto University of São Paulo, Ribeirão Preto, SP, Brazil). The natural products were denominated as AM01: 4β,5-dihydro-15-deoxy-eremantholide (MW 348); AM02: 4β,5-dihydro-2′,3′-epoxy-15-deoxy-goyazensolide (MW 362); AM03: 4β,5-dihydro-1′,2′-epoxy-15-deoxy-eremantholide (MW 364), isolated from *Eremanhus seidelii* leaves; AM04: goyazensolide (MW 360) and AM05: lychnofolide (MW 358), were isolated from *Eremanthus matogressensis* leaves and AM06: 15-deoxy-goyazenolide (MW 344), see Figure 8. Chemical compounds used as model drugs in this study (thapsigargin, THP; temozolomide, TMZ; and tunicamycin, TUN) were obtained from Sigma-Aldrich (USA). All compounds were dissolved in dimethyl sulfoxide (DMSO) as stock solutions, and diluted in culture medium as indicated.

### 4.3. Evaluation of Cell Toxicity and Cell Proliferation under Drug Treatment

The evaluation of cellular toxicity and cell proliferation was performed using the cell proliferation kit II (XTT, Roche, Mannheim, Germany), according to the manufacturer’s instructions. T98G and U87MG cell lines were seeded in 96-well plates (3 × 10^4^ and 1 × 10^4^ cells/well, for U87GM and T98G, respectively) in triplicate, and maintained in cell culture until complete adherence (24 h). Then, the culture medium was removed and replaced with DMEM containing the six sesquiterpene lactones (AM01 to AM06 at 10, 50, and 100 µM), the plates were again incubated for 24 h at 37 °C. The culture medium was removed and replaced by 100 μL of the XTT reagent. Plates were incubated again for 2–4 h and the optical density readings were taken at 450 and 650 nm. For the cell proliferation assay of compounds AM04 and AM05, U87MG and T98G were treated at the concentrations of 0 (control), 5.0, 10.0, 15.0, 20.0, 50 and 100.0 μM, for 24 h.

### 4.4. Clonogenic Assay for Sesquiterpene Lactone Compounds

Clonogenic assay or colony formation assay is an in vitro cell survival assay based on the ability of a single cell to grow into a colony [18]. U87MG and T98G cell lines were cultured until confluency. They were detached using a 0.25% trypsin/EDTA solution (Gibco, Carlsbad, CA, USA), as indicated by the manufacturer. They were seeded in a 6-well plate (300 to 1000 cells/1 mL), in triplicate, and they were allowed to adhere for 24 h. The medium was removed and replaced with 1 mL of the medium containing the compounds AM01; AM02; AM03; AM04; AM05 and AM06 at the concentration range from 0 (control), 250, 500, 1000 to 2000 nM, and the cell lines were incubated for an additional 24 h. The medium was removed and replaced with 2 mL of DMEM containing 10% FBS. Following 5–10 days of incubation, colony formation was assessed. The culture medium was removed and the plates were washed with PBS, fixed with 6% paraformaldehyde for 30 min, and then stained with 0.5% crystal violet. Excess of stain was removed by water washing. The plate was allowed to dry naturally, upside down, to avoid tarnishing the plate. The number of colonies was obtained by analyzing the digital photographs of the plates, using the program Image J or a Bio Rad image documenter (program Quantity One 4.6.1 and Gel doc XR). Plating efficiency (PE) and survival fraction (SF) were calculated through the following equations: PE = average number of control colonies formed/number of seeded cells. SF = average number of colonies formed after treatment/number of seeded cells × PE.

### 4.5. Evaluation of the Cell Membrane Permeability of AM04 (Goyazensolide) and AM05 (Lychnofolide) Sesquiterpene Lactones Isolated from Eremanthus through a Mimic Membrane Assay Formed by Monolayer of MDCK Cells

MDCK cells, 3 × 10^4^ cells/well, were plated in polycarbonate transwell inserts with a cross sectional area of 0.33 cm^2^ and 0.4 μm pores (apical compartment) in 24-well plates (Costar, Corning Incorporated, NY, USA) [19]. The medium was changed daily for 6–8 days. Twenty-four hours before the experiments, U87MG cells, 5 × 10^3^ cells/well, were plated in the basal compartment.

Cell permeability assays were performed using Hanks’ balanced salt solution (HBSS), supplemented with 10 mM HEPES and 15 mM glucose, pH 7.4, in the two compartments. AM04 (10 μM), AM05 (15 μM) and TMZ (2000 μM) were applied either to the apical compartment (insert, MDCK+) or directly into the basal compartment (MDCK-). After a 24 h incubation period, the basal compartments were treated with 0.25% trypsin/EDTA (Gibco, NY, USA) as indicated by the manufacturer. The media were collected, transferred to microtubes, and centrifuged at 300×*g* × 10 min. The cell pellet was resuspended in 0.4% Trypan blue solution (Sigma, Saint Louis, MO, USA) and the survival cells were counted in Neubauer’s chamber. The results were plotted against the untreated control. The integrity of the MDCK cell monolayer was verified by the detection of the fluorescent dye Lucifer Yellow (LY, Sigma, Saint Louis, MO, USA) in the basal compartment. LY was used at the final concentration of 4.5 μM [20]. LY fluorescence was determined on plate fluorescence reader (Enspire, Perkin Elmer, Waltham, MA, USA) at the excitation and emission wavelengths of 430 and 530 nm, respectively. The apparent permeability coefficient (Papp) of LY was calculated according to the equation: Papp = (dQ/dT)(1/AC0). The results of Papp < 0.5 × 10^6^ cells/s/cm^2^ were considered [21].

### 4.6. Wound Healing Assay

T98G and U87MG cell lines were seeded in 6 well tissue culture plates at density of 3 × 10^5^ cells/well, and were maintained in culture for 24 h to complete cell adhesion and the formation of a confluent monolayer. Monolayer cells were wounded by scratching the surface with a sterile 200 μL pipette tip to create a gap of constant width (0.4–0.5 mm). Culture medium was then immediately removed (along with any dislodged cells), and replaced with a fresh serum free culture medium, or with the medium containing treatments: AM04 (0.55 and 1.10 µM for T98G and U87MG, respectively); AM05 (0.83 and 1.66 µM for T98G and U87MG, respectively) and 2.05 µM THP, 0.5 mM TMZ and 8 µg/mL TUN for both cell lines. All scratch assays were performed in quadruplicate. Wound closure was monitored and photographed at 0 and 24 h by phase-contrast microscopy (Olympus microscope, Tokyo, Japan). The gap widths were counted in five random fields using the software ImageJ.

### 4.7. Proteome Profiler Human Apoptosis and Stress Antibody Arrays

T98G and U87MG cell lines were cultured in 75 cm^2^ flasks until reaching ≤50% confluence. The medium was removed and replaced with DMEM containing AM04 6.7 μM (U87MG) and 10 μM (T98G), 6 μM thapsigargin (THP), 1 mM temozolamide (TMZ) and 16 μg/mL tunicamycin (TUN) (for both strains, T98G and U87MG) and incubated for 24 h. The concentrations of TMZ, THP and TUN were determined by a cell proliferation assay with both cell lines and corresponded to a concentration equivalent to IC50. Then the cell lines were removed with 0.25% trypsin/EDTA solution (Gibco, NY, USA) as indicated by the manufacturer. Cells were washed with PBS 2–3 times, centrifuged at 300×*g* ×10 min. The resulting pellets were added to a cocktail of protease and phosphatases inhibitors (MS-SAFE, Sigma, Saint Louis, MO, USA) and stored at −80 °C until assayed.

Proteome profiling was performed using commercially available cell stress and apoptosis proteome profiler arrays (R&D Systems, Minneapolis, MN, USA). Expression levels of 13/26 cell stress associated proteins were evaluated in protein extracts of U87MG and T98G glioblastoma cell lines using different compounds to induce cell stress, as well as to find the expression level of 23/35 apoptosis associated proteins. To this end, the following kits were used: Proteome Profiler Human Cell Stress Array (ARY018) and Proteome Profiler Human Apoptosis Array (ARY009). The protocol and solutions used were those indicated by the manufacturer. Protein was quantified by the method of Bradford. Six experimental groups were analyzed: control, AM05, AM04, THP, TMZ and TUN, treated for 24 h. The kit consists of four membranes containing absorbed 26 or 35 duplicate antibodies, plus three reference duplicates and one negative control. The cell lysate after treatment was quantified, and equal amounts of protein (200 μg) of each sample were applied to the respective array membranes. The reading was done by biotin-streptavidin-peroxidase method, with chemiluminescence detection in the PhotoQuant4000 Photodocumentator (GE Healthcare). Densitometric analyses of the arrays were manually performed using the Studio Lite software. The pixel intensities of each spot were compared to control (untreated). The difference between pixel intensities was calculated in relation to correspondent protein in the untreated sample and expressed as percentage (%). A color grade heatmap was applied to differentiation of immunodetection for each spot.

### 4.8. Quantitative Proteome Analysis Based on Mass Spectrometry Using iTRAQ Labelling (Isobaric Tags for Relative and Absolute Quantitation)

The cell lysate obtained in the previous step was subjected to precipitation with ice-cold acetone at a ratio of 1:6. Briefly, a volume containing ~100 μg in protein was incubated with 6 volumes of ice-cold acetone at −20 °C for 16–18 h. Then, the mixture was centrifuged at 8500×*g* × 10 min, the supernatant was discarded and the residual acetone was dried on SpeedVac. The samples were resuspended in 0.1 M ammonium bicarbonate buffer, reduced with 45 mM DTT and alkylated with 100 mM iodoacetamide and hydrolyzed with endo-Lys-C/trypsin (Promega) for 24 h at 37 °C. The samples were desalted using R2 50 POROS resin, assembled in a 200 µL gel loader tip. The EndoLys-C/Trypsin peptides were dried on SpeedVac, derivatized with iTRAQ 8-plex according to manufacturer instructions and analyzed by mass spectrometry. iTRAQ samples were submitted to shotgun proteomics using UPLC-ESI-Q-TOF (Manchester, UK). Briefly, the pooled iTRAQs 8-plex (113-untreated; 114-AM04; 115-AM05; 116-THP; 117-TMZ; 118-TUN and 119-internal control) were separated in a gel loader tip loaded with POROS R2 resin in 8 fractions in acetonitrile gradient in alkaline pH (0.1% triethylamine). Each fraction was analyzed in UPLC-nanoAcquity equipped with a reverse phase column (2.1 mm × 100 mm, BHE C_18_, 1.7 µm, Waters) on line with nano-electrospray source coupled to Q-TOF-Ultima (Waters, Manchester, UK) at 35 °C, flow rate of 300 nL/min, developing a linear gradient of acetonitrile from 0.1% formic acid to 0.1% formic acid/acetonitrile for 90 min. MS and CID-MS/MS spectra were acquired in DDA mode for the 8 ions with most intensity. Spectra were processed for peak lists with MassLynx (v.4.1).

### 4.9. Protein Identification and Quantification of iTRAQ Labeled Samples

Tandem mass spectra were extracted to a Peak List using Mass Lynx 4.1. Charge state deconvolution and deisotoping were not performed. All MS/MS samples were analyzed using Mascot (Matrix Science, London, UK; version 2.4.0) and X! Tandem (The GPM, thegpm.org; version CYCLONE 2010.12.01.1). Mascot was set up to search db SwissProt_20180621 (selected for Homo sapiens, 28847 entries) assuming the digestion enzyme trypsin. X! Tandem was set up to search a subset of the SwissProt_20180621 database also assuming trypsin. Mascot and X! Tandem were searched with a fragment ion mass tolerance of 0.80 Da and a parent ion tolerance of 1.2 Da. Carbamidomethyl of cysteine and iTRAQ-8plex of lysine and the N-terminus were specified in Mascot and X! Tandem as fixed modifications. Deamidation of asparagine and glutamine, oxidation of methionine and iTRAQ8plex of tyrosine were specified in Mascot as variable modifications. Glu->pyro-Glu of the N-terminus, ammonia-loss of the N-terminus, Gln->pyro-Glu of the N-terminus, deamidation of asparagine and glutamine, oxidation of methionine, carbamidomethylation of cysteine and iTRAQ8plex of tyrosine were specified in X! Tandem as variable modifications. Scaffold (version Scaffold_4.4.1.1, Proteome Software Inc., Portland, OR, USA) was used to validate MS/MS based peptide and protein identifications. Peptide identifications were accepted if they could be established at greater than 7.0% probability to achieve an FDR less than 1.0% by the Scaffold Local FDR algorithm. Protein identifications were accepted if they could be established at greater than 99.0% probability and contained at least 2 identified peptides. Protein probabilities were assigned by the Protein Prophet algorithm [22]. Proteins that contained similar peptides and could not be differentiated based on MS/MS analysis alone were grouped to satisfy the principles of parsimony.

### 4.10. Statistical Analysis

Statistical analyses for functional assays were performed using the statistical package of GraphPad Prism software version 8.0.2 for Windows (GraphPad Software, San Diego, CA, USA, www.graphpad.com), and for transwell assay was performed paired Student t-test with a one-tailed distribution from Excel (Microsoft). Statistical analyses in proteomic analysis were performed by Scaffold version Scaffold_4.4.1.1 (Proteome Software Inc., Portland, OR, USA).

## 5. Conclusions

TMZ resistance is a major problem in the treatment of malignant brain tumors. Also, a potential problem with the use of TMZ to treat GBM patients is that the tumors may acquire TMZ resistance through alteration, not only in their expression of DNA alkylating proteins and DNA repair enzymes but also in cell signaling pathways as well. In this work we screened six low molecular weight lactone sesquiterpene as anti-proliferative agents against GBM strains and demonstrated that two of them, goyazensolide and lychnofolide, were more potent than temozolomide, according to in vitro assays for the cell lines tested here, were effective in reducing cell viability, preventing the formation of anchorage-dependent colony and were able to pass through a mimetic blood-brain barrier making them candidates for glioma therapy. Proteomic analysis revealed a number of altered proteins in expression involved in glycolytic metabolism and cellular catabolism. Globally, our results suggest that AM-04 and AM-05 may be potential candidates against TMZ-resistant gliomas. 

## Figures and Tables

**Figure 1 ijms-21-04713-f001:**
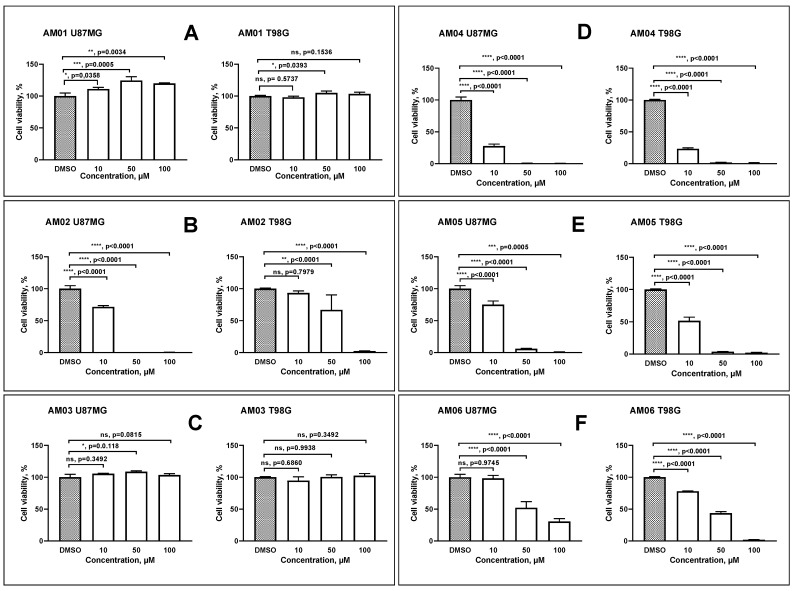
Proliferation analysis of human glioblastoma cell lines, U87MG and T98G treated with different sesquiterpene lactones. (**A**) AM01: 4β,5-dihydro-15-deoxy-eremantholide (MW 348); (**B**) AM02: 4β,5-dihydro-2′,3′-epoxy-15-deoxy-goyazensolide (MW 362); (**C**) AM03: 4β,5-dihydro-1′,2′-epoxy-15-deoxy-eremantholide (MW 364); (**D**) AM04: goyazensolide (MW 360); (**E**) AM05: lychnofolide (MW 358) and (**F**) AM06: 15-deoxy-goyazenolide (MW 344). GBM cells were treated with dimethyl sulfoxide (1% DMSO, control), 10, 50 and 100 μM of each drug. Data were represented as mean ± SEM, *n* = 3. For comparative analysis of groups of data one-way ANOVA was used, followed by Dunnett’s multiple comparisons test, performed using GraphPad Prism version 8.0.2 for Windows (GraphPad Software, San Diego, California USA, www.graphpad.com). The *p* values are presented in the figure. ns: not significant, *p* ≥ 0.05; *: significant, *p* values range between 0.01 to 0.05; **: very significant, *p* values range between 0.001 to 0.01; ***: extremely significant, *p* values range between 0.0001 to 0.001, and ****: extremely significant <0.0001.

**Figure 2 ijms-21-04713-f002:**
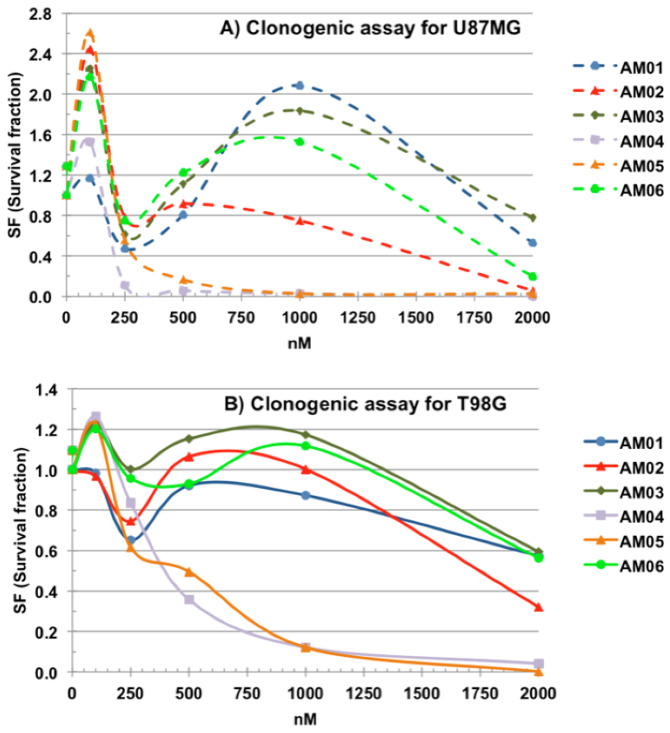
Clonogenic assay for sesquiterpene lactone compounds. (**A**) Clonogenic assay for U87MG. (**B**) Clonogenic assay for T98G. *n* = 3, mean of SF.

**Figure 3 ijms-21-04713-f003:**
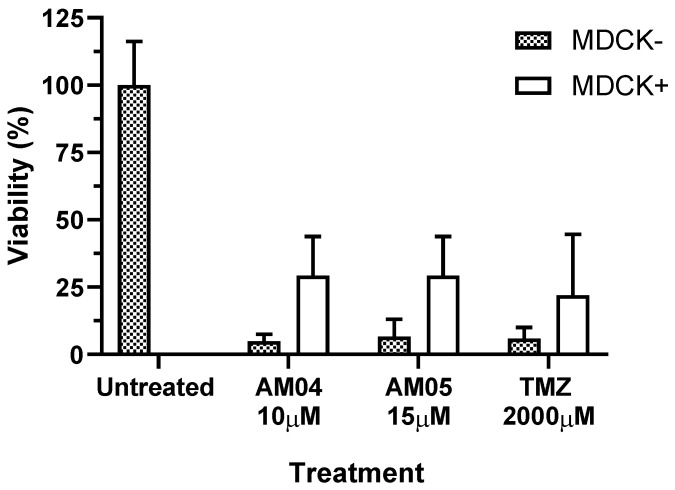
Evaluation of the cell membrane permeability of sesquiterpene lactone AM04 (goyazensolide), AM05 (lychnofolide) and temozolomide (TMZ) and untreated U87MG cells through a mimic membrane assay formed by monolayer of MDCK cells. Results are expressed in mean ± SEM, *n* = 3. For comparative analysis between untreated sample versus compounds tested with and without MDCK cell mimetic membrane, the paired Student’s *t*-test with one-tailed distribution was applied showing that all treatments were statistically significant *p* < 0.01.

**Figure 4 ijms-21-04713-f004:**
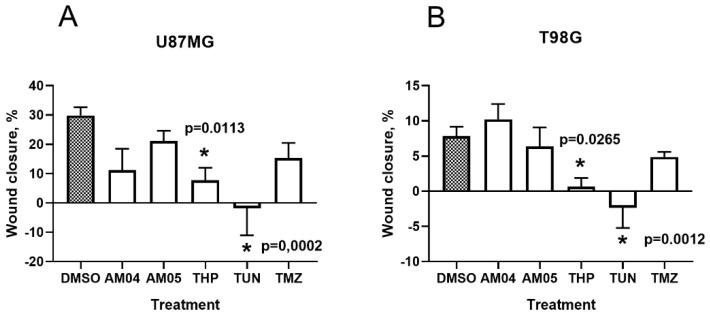
Percentage of wound closure of U87MG (**A**) and T98G (**B**) cells after the different treatments were analyzed by the wound healing assay compared to untreated control (DMSO). AM04 (0.55 and 1.10 µM for T98G and U87MG, respectively); AM05 (0.83 and 1.66 µM for T98G and U87MG, respectively) and 2.05 µM THP, 0.5 mM TMZ and 8 µg/mL TUN for both cell lines. Data were represented as mean ± SEM, *n* = 4. For comparative analysis of groups of data one-way ANOVA was used, followed by Dunnett’s multiple comparisons test, performed using GraphPad Prism version 8.0.2. for Windows, GraphPad Software, San Diego, California USA, www.graphpad.com. The *p* values are presented in the figure. *: significant, *p* values range between 0.01 to 0.05.

**Figure 5 ijms-21-04713-f005:**
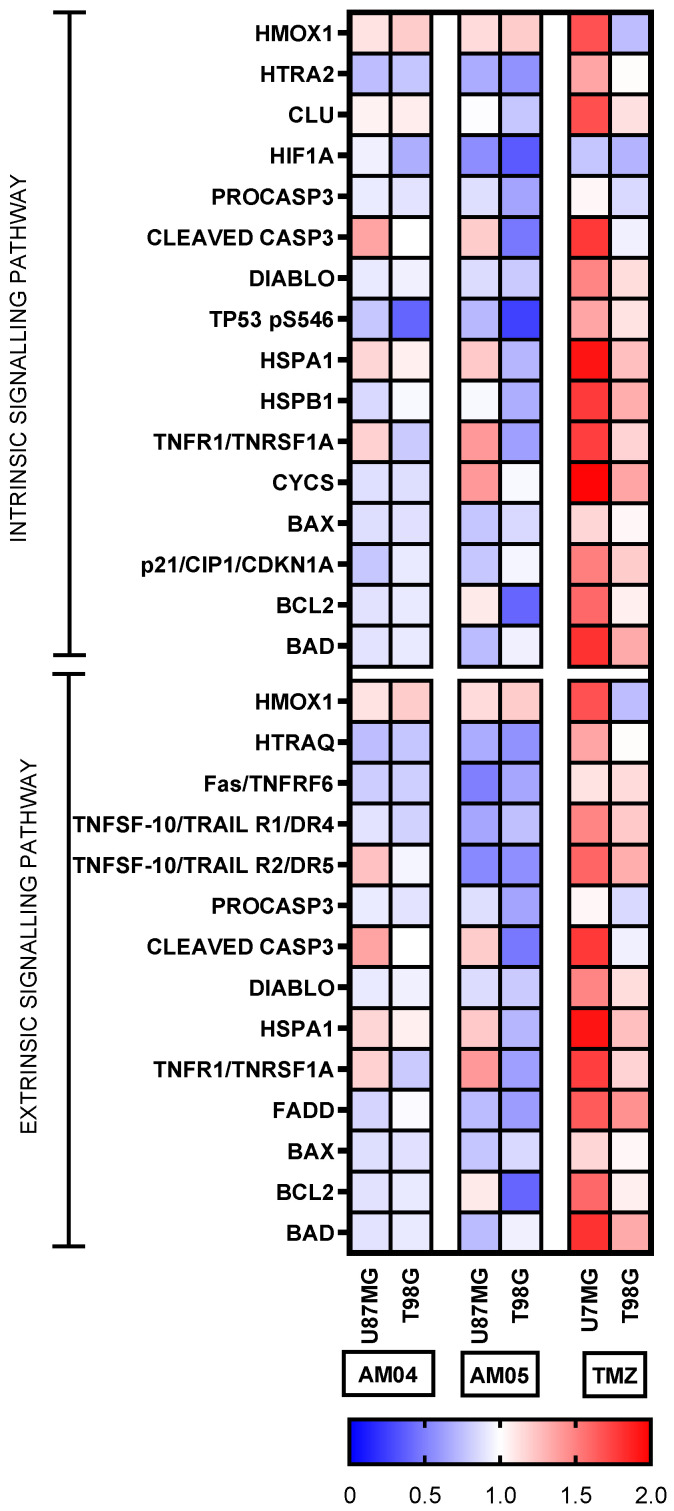
Proteome profiler of protein-related to apoptosis based on antibody array. Performed for duplicates of antibody spots for each condition.

**Figure 6 ijms-21-04713-f006:**
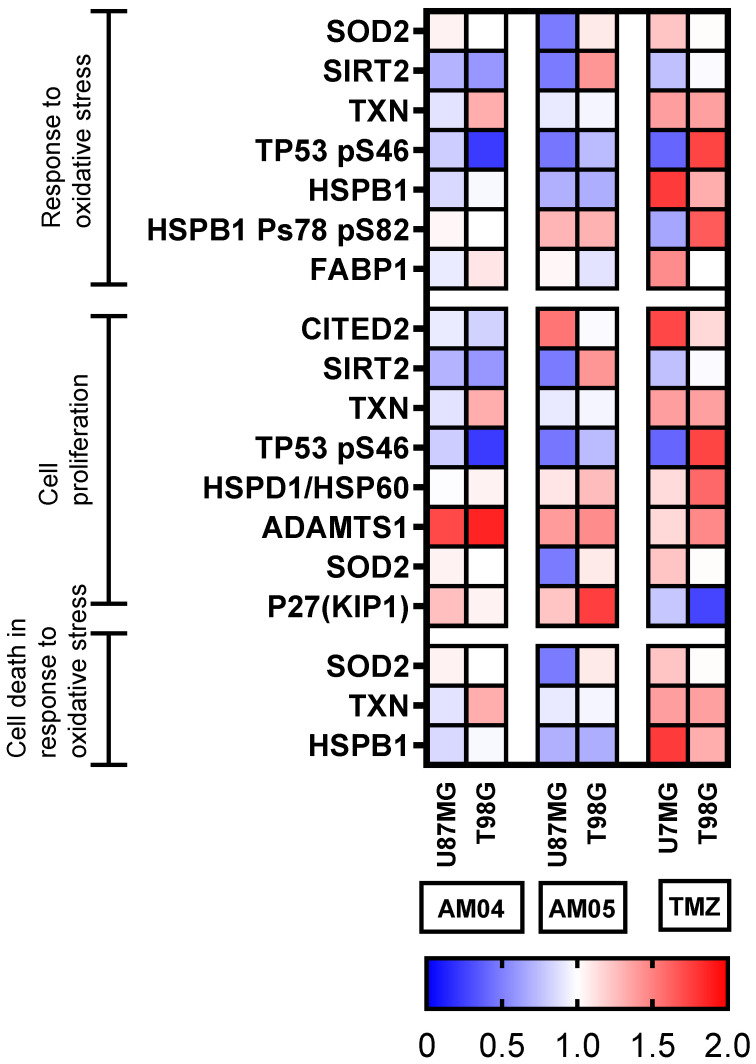
Proteome profiler of protein-related to cellular stress based on antibody array. Performed for duplicates of antibody spots for each condition.

**Figure 7 ijms-21-04713-f007:**
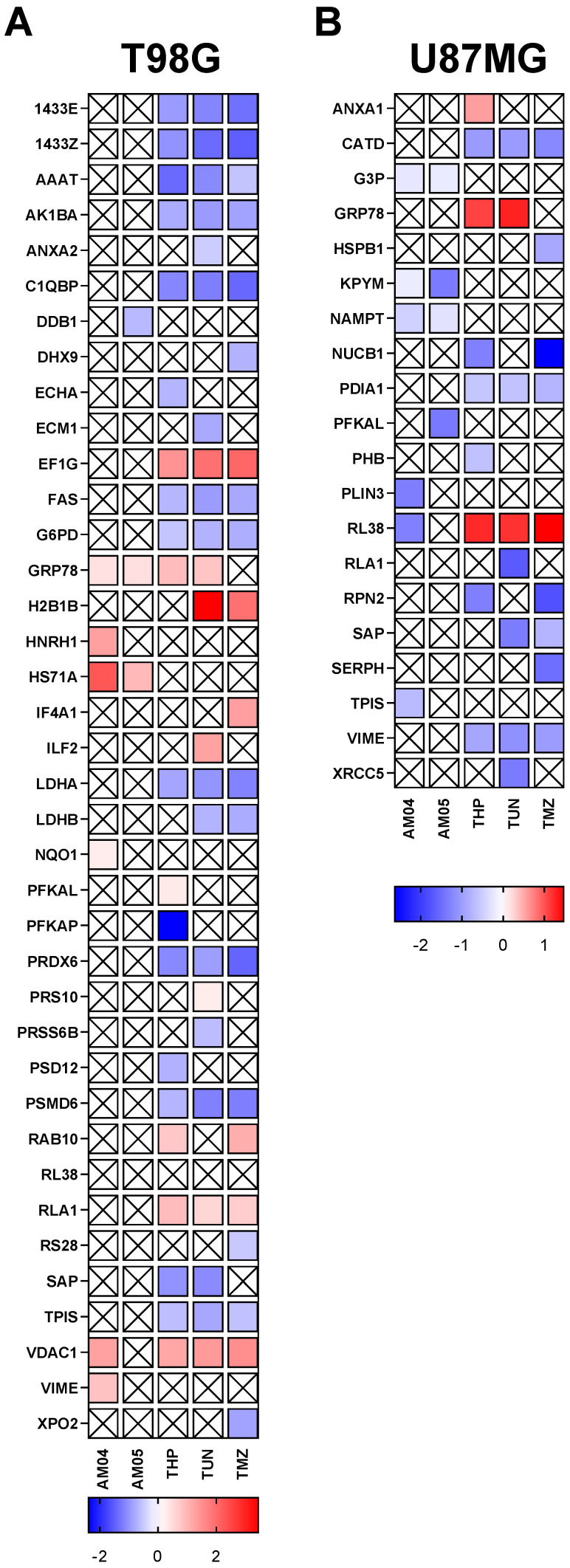
Quantitative proteomic analysis of T98G (**A**) and U87MG (**B**) under effect of several compounds obtained by shotgun bottom up proteomics after iTRAQ labelling. Protein are expressed as log2 fold change for statistically significant (*p* < 0.05, *t*-test).

**Figure 8 ijms-21-04713-f008:**
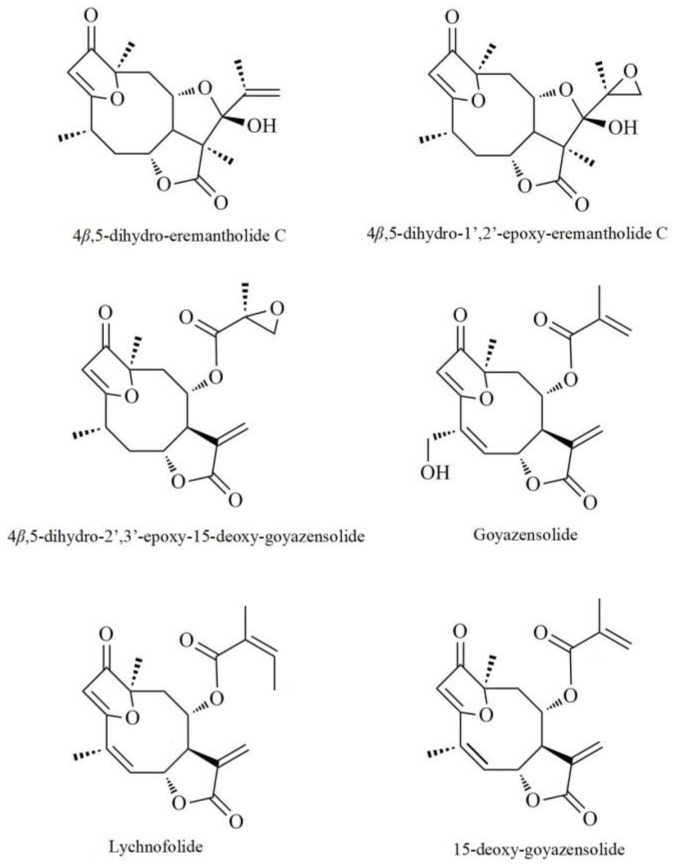
Chemical structures of sesquiterpene lactones screened in the present study, adapted from references [4] and [5].

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
