# Peer review of "Sequesterpene Lactones Isolated from a Brazilian Cerrado Plant (*Eremanthus* spp.) as Anti-Proliferative Compounds, Characterized by Functional and Proteomic Analysis, Are Candidates for New Therapeutics in Glioblastoma"

_ijms, 2020, doi:10.3390/ijms21134713_

Round 1

Reviewer 1 Report

Dear Authors,

In this work an investigation was carried out on six sesquiterpene lactone isolates from Eremanthus sp. collected from Brazilian cerrado. The anti-proliferative activity indicated that among six compounds, two compounds, AM04 (goyazensolide, MW 360) and AM05 (lychnofolide, 248 MW 358) were effective in preventing the cell proliferation of two cell types derived from glioblastoma, U87MG and T98G.

Comments:

We don't know what species is the source of the tested substances. Eremanthus sp. – this is insufficient  information. Species name should be given and plant species characteristic should be given. Such studies are not allowed without accurate identification of the plant species.

Eremanthus sp - should be: Eremanthus sp.

What tests were carried out confirming the use of parametric statistics? – information should be given in section Material and Methods. The use of t-test to analyze all data is impossible. In some cases the one-way ANOVA or two-way ANOVA and Tukey test as post-hoc test should be used. Statistical analyses should be made again in accordance with the statistical analysis procedure.

Line 35: Introduction – in the light of many results and many studied problems this part is very poor (6 sentences?) – should be strengthened. The authors did not convince me that this is an important research problem

Line 55-60: this information should be placed in the section Material and Methods

Line 57; dimethyl sulfoxide (DMSO) – this abbreviation is mentioned first time – full name should be given

Line 61-66: this was not confirmed statistically; why the statistical analysis was not made?

Line 72-73: “At these concentrations both compounds were able to reduce cell viability, T98G 72 in less than 20% and U87MG in between 20 and 35%.” – different  values ​​are given on the figure – should be corrected

Line 83: “n=3”; Line 106: “N=3” – should be unified

Line 83: Fig. 2; sequence of GBM cells should be consequently maintained as in Fig. 1

Line 86-88: this information should be placed in the section Material and Methods

Line 101: full name should be given

Line 121-122: “The results obtained for cell viability were statistically significant (p <0.01, t- test).”  - statistically significant differences are not marked in Fig. 4

Line 129: Fig. 4 - t-test is used to compare two groups. What is here compared?. Statistically significant differences are not marked. It is not clear.

Line 132: “t- test <0.01”   ?

Line 141: Fig. 5 - what statistical test was used here?

Line 73, 83, 168: p <0.05; p< 0.05; p<0.01 - standardize spaces throughout the text

Line 335, 339 – “The natural products 335 were denominated as AM01: 4β,5- dihydro- 15- deoxy- eremantholide (MW 348); AM02: 4β,5- 336 dihydro- 2',3'- epoxy- 15- deoxy- goyazensolide (MW 362); AM03: 4β,5- dihydro- 1',2'- epoxy- 15- 337 deoxy- eremantholide (MW 364); AM04: goyazensolide (MW 360); AM05: lychnofolide (MW 358) and 338 AM06: 15- deoxy- goyazenolide (MW 344 ...” – spaces in molecule names; see all text

Line 478: “Conclusions” – in the light of many results and discussion this part is very poor in information and final conclusion – should be strengthened

Line 498: References – should be prepared according to the journals's requirements. This part is very poor

The quality of figures should be improved: unification of font size, line border, line thickness, e.t.c.

Line 508: „Eremanthus” –  all Latin names should be in italics – Eremanthus

Author Response

Response to reviewer 1

Comments:

We don't know what species is the source of the tested substances. Eremanthus sp. – this is insufficient information. Species name should be given and plant species characteristic should be given. Such studies are not allowed without accurate identification of the plant species.

We added the information requested in introduction and material and methods sections.

Eremanthus sp - should be: Eremanthus sp.

We corrected in the text.

What tests were carried out confirming the use of parametric statistics? – information should be given in section Material and Methods. The use of t-test to analyze all data is impossible. In some cases the one-way ANOVA or two-way ANOVA and Tukey test as post-hoc test should be used. Statistical analyses should be made again in accordance with the statistical analysis procedure.

We corrected this point in Material and Methods and put the information on figure legends.

Line 35: Introduction – in the light of many results and many studied problems this part is very poor (6 sentences?) – should be strengthened. The authors did not convince me that this is an important research problem

Gliomas are responsible for more than 60% of all primary brain tumors. Glioblastoma multiforme, a grade IV tumor (WHO), is one of the most frequent and malignant gliomas, being TMZ the only drug of first choice for glioma chemotherapy after surgery and radiation therapy. The drugs here presented may offer a potential alternative, especially in cases of TMZ resistant gliomas.

Line 55-60: this information should be placed in the section Material and Methods

We corrected.

Line 57; dimethyl sulfoxide (DMSO) – this abbreviation is mentioned first time – full name should be given

We corrected.

Line 61-66: this was not confirmed statistically; why the statistical analysis was not made?

We corrected.

Line 72-73: “At these concentrations both compounds were able to reduce cell viability, T98G 72 in less than 20% and U87MG in between 20 and 35%.” – different values are given on the figure – should be corrected

We corrected.

Line 83: “n=3”; Line 106: “N=3” – should be unified

We unified.

Line 83: Fig. 2; sequence of GBM cells should be consequently maintained as in Fig. 1

We corrected.

Line 86-88: this information should be placed in the section Material and Methods

We moved to correct place.

Line 101: full name should be given

We corrected.

 Line 121-122: “The results obtained for cell viability were statistically significant (p <0.01, ttest).”  statistically significant differences are not marked in Fig. 4

We corrected.

Line 129: Fig. 4 - t-test is used to compare two groups. What is here compared?. Statistically significant differences are not marked. It is not clear.

We corrected.

Line 132: “t- test <0.01” ?

We corrected.

Line 141: Fig. 5 - what statistical test was used here?

We corrected.

Line 73, 83, 168: p <0.05; p< 0.05; p<0.01 -standardize spaces throughout the text

We corrected.

Line 335, 339 – “The natural products 335 were denominated as AM01: 4β,5- dihydro- 15-deoxy- eremantholide (MW 348); AM02: 4β,5-336 dihydro- 2',3'- epoxy- 15- deoxygoyazensolide (MW 362); AM03: 4β,5- dihydro-1',2'- epoxy- 15- 337 deoxy- eremantholide (MW 364); AM04: goyazensolide (MW 360); AM05: lychnofolide (MW 358) and 338 AM06: 15-deoxy- goyazenolide (MW 344 ...” – spaces in molecule names; see all text

We corrected.

Line 478: “Conclusions” – in the light of many results and discussion this part is very poor in information and final conclusion – should be strengthened

We corrected

Line 498: References – should be prepared according to the journals's requirements. This part is very poor

We corrected.

The quality of figures should be improved: unification of font size, line border, line thickness, e.t.c.

We corrected.

Line 508: „Eremanthus” – all Latin names should be in italics – Eremanthus

We corrected.

Reviewer 2 Report

English need to be reviewed throughly, e.g "tumoral adherent" line 43 page 2. Chemical structures of the six compounds need to be added. Figure 1 and 2 lacks the data of the positive control. The negative control need to be referred to by (DMSO) rather than (0). The abstract and the conclusion states that 2 com pounds were better than temozolomide but quantitative data are missing, how much better or why this is concluded is not clear.

Author Response

English need to be reviewed throughly, e.g "tumoral adherent" line 43 page 2.

We corrected.

Chemical structures of the six compounds need to be added. Figure 1 and 2 lacks the data of the positive control. The negative control need to be referred to by (DMSO) rather than (0).

We added a new Figure 8 in Material and Methods section

The abstract and the conclusion states that 2 compounds were better than temozolomide but quantitative data are missing, how much better or why this is concluded is not clear.

As mentioned in the discussion (Line 265-271). The anti-proliferative activity indicated that among six compounds, two compounds, AM04 (goyazensolide, MW 360) and AM05 (lychnofolide, MW 358) were effective in preventing the cell proliferation of two cell types derived from glioblastoma, U87MG and T98G. The presence of TP53 (mutated) in T98G and not mutated in U87MG gives them a different response to apoptotic stimuli by drugs. Furthermore, T98G is resistant to temozolomide (TMZ) by expressing an MGMT enzyme.

Reviewer 3 Report

General:

The manuscript by Izumi et al. characterizes the effect of two plant-derived compounds on glioma cell line characteristics. The authors employed functional assays to assess proliferation, viability, self-renewal and migration, as well as the potential of these drugs to cross a blood brain barrier mimicking in vitro system. Finally they performed proteome analysis via proteomics and protein arrays by immunoblotting.

Major comments:

The Reviewer feels that the way results are currently presented does not convey a clear message to the reader. Although some important aspects need to be adapted (see below), the manuscript is rather clear until the effect of AM04-05 on glioma cell viability, clonogenicity and proliferation. Then, the added-value of comparing to Thapsigargin (THP) and Tunicamycin (TUN) is ambiguous and should be better explained. The comparison to Temozolomide (TMZ) is great, however this was not performed across all presented experiments (missing for proliferation & clonogenic assay). As a major comment, across the manuscript and assays, different concentrations of AM04-05 are used. This discrepancy should be harmonized in all the assays by using for instance the same concentration employed for proteome experiments. The authors should perform all statistical tests required to support the conclusions from their study: fig 1, fig 3, fig 4. Interpretation from Fig 5 is somehow confusing. Line 135: “Wound healing assay results showed that migration of U87MG and T98G cells was significantly decreased by THP e (typo here) TUN treatments compared to control”. According to the current y axis legend “wound closure” TUN and THP seem to promote migration. Also the variability is quite big and the Reviewer has doubt concerning the statistical tests performed. The authors need to make sure that they plotted the number of experiment and not the quadruplicate (n number is missing in the legend). Proteome data suggest that AM04-05 prevent cell proliferation without inducing apoptosis (Lines 181-183). To support this data, the authors could thoroughly assess cell cycle by FACS incorporation of propidium iodide or measure Caspase 3 cleavage by immunoblotting.

Minor comments:

A reference should be provided when using MDCK as a model for BBB (line 101) as currently, the MDCK1-MDR1 cells are preferred for such application (PMID 29872378). All data not shown related to Fig 4 should be presented: MDCK viability upon TMZ and Lucifer yellow passage.

Line 120: “It should be noted that the compounds subjected to the insert with MDCK had an approximate 6- fold dilution relative to the lower chamber.” What do the authors mean? This problem is avoided if one would use empty inserts for the MDCK- condition?

The authors should also show membrane permeability on T98G cells.

Line 144: please provide the concentration of TUN in µM or mM. Line 148: “cell lysates, Proteome profiling”, typo please correct. Lines 152 & 168: the “(p<0.01)” indicated is just the cut-off used, instead please indicate the p-values for each GO, together with the official GO number when citing a GO term. Tables 1 & 2: better show as fold change instead of percentage. Indicate the color scale. Remove the “control” column. Line 177: please cite a reference supporting the anti-oxidant role of TXN.

Author Response

Response to reviewer 3

Major comments:

The Reviewer feels that the way results are currently presented does not convey a clear message to the reader. Although some important aspects need to be adapted (see below), the manuscript is rather clear until the effect of AM04-05 on glioma cell viability, clonogenicity and proliferation. Then, the added-value of comparing to Thapsigargin (THP) and Tunicamycin (TUN) is ambiguous and should be better explained. The comparison to Temozolomide (TMZ) is great, however this was not performed across all presented experiments (missing for proliferation & clonogenic assay).

The effect of TMZ is described for both cell lines

As a major comment, across the manuscript and assays, different concentrations of AM04-05 are used. This discrepancy should be harmonized in all the assays by using for instance the same concentration employed for proteome experiments. The authors should perform all statistical tests required to support the conclusions from their study: fig 1, fig 3, fig 4.

The concentration was defined for each experiment

Interpretation from Fig 5 is somehow confusing. Line 135: “Wound healing assay results showed that migration of U87MG and T98G cells was significantly decreased by THP e (typo here) TUN treatments compared to control”. According to the current y axis legend “wound closure” TUN and THP seem to promote migration. Also the variability is quite big and the Reviewer has doubt concerning the statistical tests performed.

We corrected the figure and added the statistical information

The authors need to make sure that they plotted the number of experiment and not the quadruplicate (n number is missing in the legend).

All the experiments were realized at triplicate

Proteome data suggest that AM04-05 prevent cell proliferation without inducing apoptosis (Lines 181-183). To support this data, by FACS incorporation of propidium iodide or measure Caspase 3 cleavage by immunoblotting.

We performed the western blotting analysis for Caspase-3, and no caspase cleavage was detected

Minor comments:

A reference should be provided when using MDCK as a model for BBB (line 101) as currently, the MDCK1-MDR1 cells are preferred for such application (PMID 29872378). All data not shown related to Fig 4 should be presented: MDCK viability upon TMZ and Lucifer yellow passage.

We corrected.

Line 120: “It should be noted that the compounds subjected to the insert with MDCK had an approximate 6- fold dilution relative to the lower chamber.” What do the authors mean?

This problem is avoided if one would use empty inserts for the MDCK- condition? The authors should also show membrane permeability on T98G cells.

Text and Figure were altered in order to better explain the assay

Line 144: please provide the concentration of TUN in μM or mM.

Tunicamycin used in this study is a mixture of homologous antibiotics produced by Streptomyces sp which contain uracil, N-acetyl glycosamine, an 11-carbon aminodialdose called tunicamine, and a fatty acid linked to the amino group. There are at least 10 homologs, the main components being A, B, C, and D. The homologs differ in their fatty acid components, which vary the chain length. For this reason, we used mass/vol instead of molar concentration.

Line 148: “cell lysates, Proteome profiling”, typo please correct.

We corrected.

Lines 152 & 168: the “(p<0.01)” indicated is just the cut-off used, instead please indicate the p values for each GO, together with the official GO number when citing a GO term.

The paragraph was removed from the text, once the array already specifies the biological process in  which the proteins are involved

Tables 1 & 2: better show as fold change instead of percentage. Indicate the color scale. Remove the “control” column.

We changed both for a heat map figure.

Line 177: please cite a reference supporting the anti-oxidant role of TXN.

The following reference was added to the manuscript: Lu J, Holmgren A. “The thioredoxin antioxidant system”. Free Radic Biol Med. 2014 Jan;66:75-87

Round 2

Reviewer 1 Report

Dear Editor,

Comments

Line 3 – in this moment we know that we have 2 plant species – in title should be: “

 Eremanthus spp.

Explanation:

  1. – species

spp. – species plural

Figure 1 – viability (%) can be over 100%?

Which is A, B, C … - in this version lack on figure

Why on the all figures axis OY is in the range 0-150%? viability can be over 100%?

Figure 4 - showing differences with asterisks is incorrect. In the case of E and F anover test sholuld be used.

Where are the results of ANOVA? In the case of two-way ANOVA the interaction of the two factors is analysed. Where is the description

Line 94 – Eremanthus sp

Line 28 and all text -  is “Eremanthus sp” but should be “Eremanthus spp.”

Line 529-532

4.10. Statistical Analysis

Statistical analyses were performed using GraphPad Prism software v. 8.0.2. All statistical analyses in proteomic analysis were performed by Scaffold (version Scaffold_4.4.1.1, Proteome Software Inc., Portland, OR.”

Statistical methods are not described.

The basic problem is still the lack of proper statistical tools. The authors should be use the help of proffesional statistics. Therefore, a review of the work is nounnecessary at this time.

Author Response

The authors would like to deeply thank your invaluable analysis of the text, as well as the questions raised during the manuscript review process.

Reviewer 1

Comment (1):

Line 3 – in this moment we know that we have 2 plant species – in title should be: “ Eremanthus spp.

Explanation:

– species

spp. – species plural

Answer: We corrected in the text

Comment (2):

Figure 1 – viability (%) can be over 100%?

Answer: Cell viability is above 100% to fit the standard deviation bars of the controls. This approach is used even in other articles by this publisher.

Comment (3):

Which is A, B, C … - in this version lack on figure

Answer: It was corrected. See lines…….

Comment (4):

Why on the all figures axis OY is in the range 0-150%? viability can be over 100%?

Answer: The y-axis ranges were kept similar for comparison. Cell viability is above 100% to fit the standard deviation bars of the controls.

Comment (5):

Figure 4 - showing differences with asterisks is incorrect. In the case of E and F another test should be used.

Answer: We correct the figure. Now it corresponds to figure3.  See lines…….

Comment (6):

Where are the results of ANOVA? In the case of two-way ANOVA the interaction of the two factors is analysed. Where is the description

Answer: We excluded this figure.

Comment (7):

Line 94 – Eremanthus sp

Answer: Corrected in the manuscript

Comment (8):

Line 28 and all text -  is “Eremanthus sp” but should be “Eremanthus spp.”

Answer: Corrected in the manuscript.

Comment (8):

Line 529-532

“ 4.10. Statistical Analysis

Statistical analyses were performed using GraphPad Prism software v. 8.0.2. All statistical analyses in proteomic analysis were performed by Scaffold (version Scaffold_4.4.1.1, Proteome Software Inc., Portland, OR.”

Statistical methods are not described.

Answer: Each statistical test is explained in the corresponding figure.

Comment (8):

The basic problem is still the lack of proper statistical tools. The authors should be use the help of proffesional statistics. Therefore, a review of the work is nounnecessary at this time.

Answer: The statistical data were revised and, when necessary, redone.

Reviewer 2 Report

The manuscript is good for publication

Author Response

The authors would like to deeply thank your invaluable analysis of the text, as well as the questions raised during the manuscript review process.

Reviewer 3 Report

General:

In the revised manuscript by Izumi et al. the authors have included new results and clarifications which have strengthened their findings. However, to the Reviewer opinion additional modification are still required.

Major comments:

  1. The effect of AM04-05 and TMZ on MDCK viability was tested upon two different MDCK cell confluency. The rationale to include this data was to provide the basis for the use of AM04-05 as drugs capable of crossing the BBB to target glioma cells. However, these results, initially present in the first version of the manuscript (old Figure 4), are now absent from the revised article. Besides, the material and methods still refer to these currently absent data. Therefore, the author should include this data or modify their discussion & conclusion accordingly. Of course, if the data are included, the inhibitor’ concentrations used should be concordant with the other assays (eg viability of glioma cells across the MDCK barrier). Finally, the cell density used should be congruent between the figures and methods sections, either use as cell/well or cell/cm2.
  2. There are inconsistencies in the data between Figures 1 & 2. For Figure 1, with 10µM of AM04 the U87MG viability reaches 25% whereas on Figure 2, the same concentration gives now 75% of viability. Same goes for 50µM of AM04 on U87MG, and 10µM of AM05 on T98G. How the authors explain these discrepancies? 

Minor comments:

  1. On Figure 2, why Sidak’s comparison was used instead of Dunnett as for all the other figures?
  2. The authors should correct all the typos mentioned during the first revision phase. Especially when they do state in their response letter that they corrected them. See for instance line 146 “THP e TUN treatments compared to control”; and line 161 “cell lysates, Proteome profiling”. A typo is present in supplementary figures 1-4 “controle”. Please correct.
  3. Lines 168-169: “However, treatment with AM04 and AM05 did not 168 significantly induce apoptosis in GBM cells.” The authors should clarify the statistics. According to supplementary figures 1-4, the antibody arrays from main figures 6 & 7 were performed once (n=1) per condition. This information should be clearly mentioned in the manuscript (eg in figure legends).

Author Response

The authors would like to deeply thank your invaluable analysis of the text, as well as the questions raised during the manuscript review process.

Reviewer 3

In the revised manuscript by Izumi et al. the authors have included new results and clarifications which have strengthened their findings. However, to the Reviewer opinion additional modification are still required.

Major comments:

 Comment (1):

The effect of AM04-05 and TMZ on MDCK viability was tested upon two different MDCK cell confluency. The rationale to include this data was to provide the basis for the use of AM04-05 as drugs capable of crossing the BBB to target glioma cells. However, these results, initially present in the first version of the manuscript (old Figure 4), are now absent from the revised article. Besides, the material and methods still refer to these currently absent data. Therefore, the author should include this data or modify their discussion & conclusion accordingly. Of course, if the data are included, the inhibitor’ concentrations used should be concordant with the other assays (eg viability of glioma cells across the MDCK barrier). Finally, the cell density used should be congruent between the figures and methods sections, either use as cell/well or cell/cm2.

Answer:

We apologize for this mistake. Please, see the text in new revised manuscript, lines ………..156 to 192. The old figure 4 was re-introduced as figure 3, since this result is one of most important finding on this manuscript. Many drugs against glioma have been failure due to inefficient capacity to cross the BBB. Of course, there will be a long way to validate the AM04 and AM05 to the use on clinic.

Figure mentioned above was moved to supplemental material (supplemental figure S1), line 181.

About cell density used, we corrected to number of cells/well, lines…… 564 and 567

Comment (2):

There are inconsistencies in the data between Figures 1 & 2. For Figure 1, with 10µM of AM04 the U87MG viability reaches 25% whereas on Figure 2, the same concentration gives now 75% of viability. Same goes for 50µM of AM04 on U87MG, and 10µM of AM05 on T98G. How the authors explain these discrepancies?

Answer: We accept the comment and agree that the data is not relevant and we removed this figure and data from the manuscript. We apologize for the redundancy of cell viability, but we could not explain this difference of cell viability as most of initial experimental data was exploratory and introduce as mistake.

Comment (3):

Minor comments:

On Figure 2, why Sidak’s comparison was used instead of Dunnett as for all the other figures?

Answer:  Figure 2 was excluded, due inconsistency detected by reviewer and we apologize for this mistake.

Comment (4):

The authors should correct all the typos mentioned during the first revision phase. Especially when they do state in their response letter that they corrected them. See for instance line 146 “THP e TUN treatments compared to control”; and line 161 “cell lysates, Proteome profiling”. A typo is present in supplementary figures 1-4 “controle”. Please correct.

Answer:  These mistakes were observed and corrected.

Comment (5):

Lines 168-169: “However, treatment with AM04 and AM05 did not significantly induce apoptosis in GBM cells.” The authors should clarify the statistics. According to supplementary figures 1-4, the antibody arrays from main figures 6 & 7 were performed once (n=1) per condition. This information should be clearly mentioned in the manuscript (eg in figure legends).

Answer: The information will be made clear in the legend. We introduce “Performed for duplicates of antibody spots for each condition.”

Round 3

Reviewer 1 Report

Dear Authors,

I accept the article in present form.

Reviewer 3 Report

The authors have addressed all my comments and I have no further questions regarding the manuscript by Izumi et al.